# A 47-Year Comparison of Lower Body Muscular Power in Spanish Boys: A Short Report

**DOI:** 10.3390/jfmk5030064

**Published:** 2020-08-20

**Authors:** Iván Chulvi-Medrano, Manuel Pombo, Miguel Ángel Saavedra-García, Tamara Rial Rebullido, Avery D Faigenbaum

**Affiliations:** 1Sport Performance and Physical Fitness Research Group (UIRFIDE), Department of Physical and Sports Education, Faculty of Physical Activity and Sport Sciences, University of Valencia, 46010 Valencia, Spain; 2Department of Physical Education and Sport, University of Coruña, 15179 A Coruña, Spain; pombo@udc.es; 3Group of Research in Sport Science (INCIDE), Department of Physical Education and Sport, University of Coruña, 15179 A Coruña, Spain; miguel.saavedra@udc.es; 4Exercise & Women’s Health, Newtown, CT 18940, USA; rialtamara@gmail.com; 5Department of Health and Exercise Science, The College of New Jersey, Ewing, NJ 08628, USA; faigenba@tcnj.edu

**Keywords:** pediatric dynapenia, children, resistance training

## Abstract

Much of the evidence examining temporal trends in fitness among youth has found a decrease in measures of muscular strength and muscular power over recent decades. The aim of this study was to examine trends in lower body muscular power in Spanish boys over 47 years. In 1969 140 boys (10–11 years; body mass index = 19.24, SD = 2.91 kg/m^2^) and in 2016, 113 boys (10–11 years; body mass index = 19.20, SD = 3.15 kg/m^2^) were recruited. Lower body power was assessed using the vertical jump (VJ) and standing long jump (SLJ) tests. Significant differences and a large effect size were shown between groups in the SLJ (*p* = 0.001; d = 0.94) and the VJ (*p* = 0.001; d = 0.66). SLJ data in 1969 were higher (1.52 m, SD = 0.19) when compared to the 2016 data (1.34 m, SD = 0.18). The VJ performance of the 1969 sample was also higher (25.95 cm; SD = 6.58) than the 2016 sample (21.56 cm; SD = 4.72). SLJ and VJ performance of the 2016 group decreased 11.8% and 16.9%, respectively. There were no significant differences between groups in body mass index. The results indicate a secular decline in lower body muscular power in 10–11-year-old Spanish boys with no significant changes in body mass index over the 47-year study period.

## 1. Introduction

Low levels of muscular fitness (i.e., muscular strength, muscular power and local muscular endurance) in children and adolescents are associated with poor motor competence, functional limitations and adverse health outcomes [1,2]. Recent findings indicate that measures of muscular strength and power in modern-day youths are lower than in previous generations [3,4,5,6]. Sandercock and Cohen reported a decline in muscular fitness (bent-arm hang, sit-ups and handgrip) using allometric equations in 10-year-old English children from 1998 to 2014, and noted this trend was independent of secular changes in body size [5]. A similar trend in muscular fitness was observed in Spanish adolescents between 2001–2002 and 2006–2007 [4], and in an international sample of children and adolescents between 1964 and 2017 [7].

Lower levels of muscular strength and power in modern day youths appear to be consequent to lifestyles characterized by reduced physical activity and increased sedentary behavior [3,5,8]. Importantly, muscular strength and fundamental movement skill proficiency are considered foundational for ongoing participation in physical activity across the lifespan [8,9]. Therefore, it is critical to examine temporal trends in muscular fitness in youth due to the far-reaching implications for disease prevention and health promotion. A recent meta-analysis concluded that poor muscular fitness was associated with lower levels of bone mineral density and self-esteem, as well as higher levels of body fat and cardiometabolic risk [1]. In support of these observations, lower handgrip levels in youth with obesity have been associated with increased cardiometabolic risk [10]. Further, low levels of performance on selected measures of muscular fitness, including the handgrip, push-up and long jump early in life have been found to be associated with an increased risk of metabolic syndrome later in life [11]. Specifically, the long jump is a field test commonly used in youths as a general measure of lower body muscular fitness [12].

Temporal trends of muscular fitness performance in youth can be used to inform public health policies about youths’ physical activity and health [13]. However, there are a limited number of studies examining trends in lower body muscular fitness in youths over recent decades. Additional data are needed to fill this research gap. The aim of this study was to describe temporal trends in lower body muscular power in Spanish boys over a 47-year time period from 1969 to 2016. We hypothesized that contemporary trends towards decreasing levels of muscular strength and muscular power shown in previous studies will be similar in Spanish children in 2016 as compared to Spanish children in 1969.

## 2. Materials and Methods

Data were collected from two separate cross-sectional samples from the same school in Galicia, Spain. The sample consisted of the total boys (10 to 11 years of age) enrolled in the school during both years. Participants included 140 boys (10–11 years; body mass index = 19.24, SD = 2.91 kg/m^2^) in 1969 and 113 boys (10–11 years; body mass index = 19.20, SD = 3.15 kg/m^2^) in 2016. The University Da Coruña Ethics Committee approved this research study and parents and participants were informed about experimental procedures and provided parental permission and child assent, respectively.

Body mass index (BMI) was calculated as body mass measured on an analogic scale to the nearest 0.1 kg divided by height measured on a stadiometer to the nearest 0.5 cm squared. Lower body power was assessed using the vertical jump (VJ) and standing long jump (SLJ) tests following standardized procedures [14]. In the SLJ the participant stood with both feet just behind the starting line on a marked floor. The distance between the starting line and the back edge of the participant’s heel was measured after each jump. For the VJ test, participants were instructed to jump as high as possible and mark the wall with chalk on their fingers. The vertical jump was calculated by subtracting a participant’s standing reach height from the maximal jump height. Participants were permitted to perform a countermovement prior to jumping vertically or horizontally. The test order was randomized, and all participants performed 3 trials per test. Participants were allowed to rest for 1 min between trials and for 3 min between the VJ and SLJ tests. The best score from each test was used for data analysis. In 2016, the same testing protocols and procedures were followed as in 1969. The VJ and SLJ were part of the Spanish physical education curricula during the study period and therefore all participants had 2 familiarization sessions with these tests.

A Kolmogorov–Smirnov test was conducted to determine whether the data met the assumptions of normality of distribution. When the data met the normal distribution, two-tailed *t*-tests were applied for normally distributed data and the Mann–Whitney test was used for non-normal data to determine differences between cohorts. Significance was stablished at *p* < 0.05. D-Cohen effect size (ES) was calculated using the recommended Equation (1) [15].
(1)$$\mathrm{ES}\;=\;\frac{\mathrm{mean}\;\mathrm{of}\;\mathrm{the}\;\mathrm{experimental}\;\mathrm{group}\;-\;\mathrm{mean}\;\mathrm{of}\;\mathrm{the}\;\mathrm{control}\;\mathrm{group}}{\;\mathrm{standard}\;\mathrm{deviation}\;\mathrm{of}\;\mathrm{the}\;\mathrm{control}\;\mathrm{group}}$$

## 3. Results

In 1969, the sample mean age was 10.53 (0.50) years-old (min 10- max 11-years-old) and the median was 11-years-old. In 2016, the sample mean age was 10.43 (0.49) years-old (min 10- max 11-years-old) and the median was 10-years-old. Characteristic outcomes in the different cohorts are presented in Table 1.

## 4. Discussion

Significant differences and a large effect size were found in SLJ and VJ performance from 1969 to 2016. Over this 47-year period, SLJ performance decreased 11.8% and VJ decreased 16.9%. These results indicate a declining trend in lower body muscular power in 10–11-year-old Spanish boys with no significant changes in BMI. Our findings show a greater decline in SLJ than in previous research. We observed an 11.8% decline in SLJ between 1969 and 2016, whereas Moliner-Urdiales and colleagues found a decline in SLJ performance of 4.8% in 12.5- to 17.5-year-old Spanish adolescents assessed between 2001–2002 and 2006–2007 [4]. Similarly, Hardy and colleagues reported a decline in SLJ in youth between 1985 and 2015, with 10-year-old boys decreasing SLJ performance by 4.8% [13]. These trends in measures of lower body muscular strength and muscular power in youth may be explained by declining levels of regular participation in physical education, outdoor active play, and sport activities during the respective study periods [16]. Of interest, Tomkinson and colleagues quantified global changes in anaerobic fitness in more than 20 million youths and reported improvements in performance from 1958 to 1982, followed by a plateau and eventual decline in anaerobic test performance until 2003 [17]. This observed decline in anaerobic performance is consistent with our observed temporal trends in musculoskeletal fitness. Similarly, Kaster and colleagues evaluated temporal trends in sit-up performance in almost 10 million children and adolescents and reported large international improvements from 1964 to 2000 before then stabilizing near zero until 2010 before declining [7]. Of note, national trends in sit-up performance were strongly and positively associated with trends in vigorous physical activity, with countries with the largest improvements in sit-up performance reporting the largest increases in vigorous physical activity [7].

Another outcome of interest is the BMI. Our findings indicate that there was no significant difference in BMI between 1969 and 2016. However, the 2016 boys were shorter with a lower body mass than in 1969. These differences could be attributed to a younger age of development. While BMI in our study showed no significant differences over a 47-year period, other reports observed an increase in BMI over a period of 30 years [13]. Our findings are not consistent with others who reported that today’s youths are taller and heavier than previous generations [13]. Of interest, the Spanish physical education curricula in the 1960s, 1970s and 1980s typically included more strength- and skill-building physical exercises, as compared to the more recent focus on aerobic games and activities. As such, trends in BMI and body composition must be viewed in light of the type and intensity of exercise performed, as well as the age, sex and biological maturation of the study participants.

We found a decrease in lower body power independent of changes in body size and BMI over the study period. The same downward trend in muscular fitness independent of secular changes in body size and BMI was also observed in 10-year-old English children, as reported by Sandercock and Cohen [5]. While our 1969 sample was heavier and taller than our 2016 sample, with no significant differences in BMI, English children in the Sandercock and Cohen study were taller and heavier in 2014 than in 2008 and 1998, with no significant differences in BMI [5]. Collectively, these results suggest that changes in muscular fitness over time may not be associated with secular changes in BMI and body size. It is possible that other factors, such as trends in quality and quantity of physical activity engagement, may contribute to the observed decline in muscular fitness in modern day youths. This downward trend in muscular fitness in modern day youths has also been reported in children and adolescents from other countries [3]. It should be noted that, without regular opportunities to engage regularly in strength-building exercises, today’s youths may be less likely to attain the adequate levels of muscular fitness that are needed for ongoing participation in MVPA [3,18]. Since low levels of muscular strength and power early in life are risk factors for pediatric dynapenia and associated health-related concerns [19], the SLJ has been suggested as a valid general index for assessing muscular fitness in youths [12].

Our results show that the 2016 cohort was shorter and their SLJ performance was significantly lower than in 1969. In young adults, SLJ performance has been found to be influenced by several factors, including anthropometrics [20]. Taller subjects have a higher center of gravity and longer leg lever that can produce greater mechanical jump forces. It has been reported that femur length has a significant influence on VJ performance [21]. In 10- to 12-year-old children, a positive moderate correlation between femur length and SJL, and a weak positive correlation between standing height and SLJ, were reported [22]. Given these observations, participant’s anthropometrics may affect the take-off angle during the SLJ and, consequently, the jump distance.

Our study has several limitations that should be considered when interpreting the data. Notably, we included a relatively small sample of boys from Galicia, a region of Spain. Therefore, our data are not representative of all Spanish youths. Additionally, we did not assess the biologic maturation of the participants and we did not measure their current levels of physical activity with validated questionnaires. Additionally, we did not include the periodic testing of muscular fitness during the 47-year study period and, therefore, changes in performance during selected periods of time cannot be analyzed. The assessment of body composition in order to differentiate between lean body mass and fat mass was not performed. Additional research is warranted to address the changes in muscular phenotype (e.g., muscular strength and power) in girls and boys, while controlling for confounding variables, such as exercise participation and training history. Finally, it has been reported that sociodemographic variables [13] and the trends towards unhealthy eating habits [23] may influence fitness performance in youths; however, we did not assess these variables.

## 5. Conclusions

To the best of our knowledge no previous studies have examined temporal trends in muscular fitness in youths over a 47-year period. Our novel findings are consistent with other reports from western societies that highlight declines in measures of muscular strength and muscular power in modern day children and adolescents. To alter the current trajectory towards lower levels of muscular fitness in children and adolescents, developmentally appropriate interventions that target neuromuscular deficiencies and enhance muscular strength and power are needed to avert the troubling consequences of pediatric dynapenia.

## Figures and Tables

**Table 1 jfmk-05-00064-t001:** Participant demographics and lower body muscle power over a 47-year-period in Spanish boys.

Variables	Sample 1 1969 *n*= 140 Mean (SD)	Sample 2 2016 *n*= 113 Mean (SD)	Δ	*p*-Value	Cohen’s *d*
Height (cm)	147.06 (6.21)	140.25 (6.17)	−4.6	0.001	1.18
Mass (kg)	41.68 (6.21)	38.01(8.08)	−8.8	0.001	0.50
BMI ^1^ (kg/m^2^)	19.24 (2.91)	19.20 (3.15)	−0.20%	0.473	0.01
SLJ ^1^ (m)	1.52 (0.19)	1.34 (0.8)	−11.8%	0.001	0.94
VJ ^1^ (vm)	25.95 (6.58)	21.56 (4.72)	−16.9%	0.001	0.66

^1^ BMI: Body Mass Index; SLJ: Standing Long Jump; VJ: Vertical Jump.

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
