# Peer review of "A 47-Year Comparison of Lower Body Muscular Power in Spanish Boys: A Short Report"

_jfmk, 2020, doi:10.3390/jfmk5030064_

Round 1

Reviewer 1 Report

The authors address a problem of increasing concern for the scientific community. With the development of the technology and the change in lifestyle, a decrease in the practice of physical exercise by children and young people is observed almost everywhere in the world. This problem has had a greater incidence in the Western world, resulting in the decrease of physical fitness of young people. This situation in the medium / long term has an impact on the health of the populations, being the prelude to the increased cardiovascular diseases, obesity and loss of muscle mass.
In the present case, the authors study children from a Spanish region, which corroborates results obtained by other researchers.

The methodology used seems to be appropriate to answer the presented hypothesis and the conclusions are supported by the results.

I have positive comments about the report and feel that it can contribute to the literature, but there are also a few issues the authors need to address.

Pp 3-Line 98 to 103 - I do not understand, in terms of strength children are decreasing however the anaerobic capabilities are increasing. It is not a contra sense? How do you explain this behavior?

Pp3, line 110 to 114 - You collected data for a region and then generalized it to the whole of Spain. Is this region not very particular since the Spanish population is currently taller and heavier than the 1960s?

Author Response

Dear Editor, the authors appreciate the comments for our manuscript jfmk-880289. Our manuscript has been improved with the reviewer’s comments. Changes in our manuscript have been highlighted in red in order to facilitate the location of the suggested changes. We have addressed all reviewer’s concerns and included a point-by-point response below:

Responses to the reviewer 1

Pp 3-Line 98 to 103 - I do not understand, in terms of strength children are decreasing however the anaerobic capabilities are increasing. It is not a contra sense? How do you explain this behavior?

We cited data from Tomkinson et al. (17) research where global changes in anaerobic performance in youth was assessed from 1958 to 2003. In the review from Tomkinson et al, they reported early improvements in anaerobic performance from 1958 to 1982 and then a plateau and eventual decline in performance until 2003. This observed decline in anaerobic performance is consistent with our observed temporal trends in musculoskeletal fitness.

Pp3, line 110 to 114 - You collected data for a region and then generalized it to the whole of Spain. Is this region not very particular since the Spanish population is currently taller and heavier than the 1960s?

We noted this concern in the limitations of our paper such that our findings may not be generalizable to the Spanish population.

Reviewer 2 Report

Dear Journal of Functional Morphology and Kinesiology Staff,
thank you very much for the possibility to serve as a Reviewer in a prestigious periodical like Journal of Functional Morphology and Kinesiology.

About this paper titled “A 47-year comparison of lower body muscular power 2 in Spanish boys. A short report”, the contents and the rhetoric by which it was handled are appreciable and the paper is well organized.

Brief Overview:

The objective of this study is to assess lower body muscular power in Spanish boys over 47 years. The sample is composed by two groups: 140 boys tested in 1969 and 113 boys tested in 2016. Assessments performed were: vertical jump (VJ) and standing long jump (SLJ) tests. Data indicate that performance of the 1969 sample was higher in both tests.

General comment:

This paper examining an interesting topic: muscular power and strength in youth, comparing two different boys’ generation. Although, the matter of the study is quite original, some questions require clarification in order to improve the quality of the manuscript. I have listed below specific comments to the authors.

Abstract:

I suggest to check if JFMK accepts the abstract structure that you use.

Material and Methods:

Lines 67: “Data were collected from two separate cross-sectional study”. I suggest to cite the first study (1969). Or these data were recruited for this study?

Lines 73-74: Instruments used to collect data are the same for both years (1969 – 2016)?

Lines 73-78: Was performed a test familiarization in both testing sessions? This aspect add to sedentary behavior could influence the results?

Line 76: I suggest to provide a more details about testing procedure (tools used for example).

I suggest to provide the sample size calculation and formula used.

Lines 78-80: I suggest to change “were used “ in “was applied”.

Statistic:

I suggest to carry out the correlation between anthropometric measures and jump data, and the minimum detectable change (MDC). These results could improve discussion and conclusions

Results:

Lines 84-85: Why the sample size is different? I suggest to motivate this.

Line 92: I suggest to change “Secular”

Discussion:

Lines 111-112: “However, the 2016 boys were shorter with a lower BMI”; I think BMI is incorrect. Do you mean mass?

Lines 127-128: Did you collect data about physical activities such as IPAQ, GPAQ, questionnaires? I think these data could give an important contribution to better explain the findings.

Lines 137-145: I think that another study limit is that lean and fat mass were not assessed. I suggest to add this thought to limit session.

Finally, it’s possible that height influenced standing long jump test? I think that aspect should be rationalize/elucidate in the discussion.

Table 1.

I suggest to provide a title table.

Author Response

Dear Editor, the authors appreciate the comments for our manuscript jfmk-880289. Our manuscript has been improved with the reviewer’s comments. Changes in our manuscript have been highlighted in red in order to facilitate the location of the suggested changes. We have addressed all reviewer’s concerns and included a point-by-point response below:

Responses to the reviewer 2

Abstract:

I suggest to check if JFMK accepts the abstract structure that you use.

Thank you. We checked JFMK abstract instructions and we have edited the headings and to ensure it follows JFMK guidelines.

Material and Methods:

Lines 67: “Data were collected from two separate cross-sectional study”. I suggest to cite the first study (1969). Or these data were recruited for this study?

We made the following edit to clarify the text. “Data were collected from two separate cross-sectional samples from the same school in Galicia, Spain.”

Lines 73-74: Instruments used to collect data are the same for both years (1969 – 2016)?

Thank you. We made the following edit to clarify the text “In 2016 the same testing protocols and procedures were followed as in 1969.”

Lines 73-78: Was performed a test familiarization in both testing sessions? This aspect add to sedentary behavior could influence the results?

The VJ and SLJ were part of the Spanish physical education curricula during the study period and therefore all participants had 2 familiarization sessions with these tests.

Line 76: I suggest to provide a more details about testing procedure (tools used for example).

Thank you. We edited the text for clarity: “In the SLJ the participant stood with both feet just behind the starting line on a marked floor. Distance between the starting line and the back edge of the participant’s heel was measured after each jump.  For the VJ test, participants were instructed to jump as high as possible and mark the wall with chalk on their fingers. The vertical jump was calculated by subtracting a participant’s standing reach height from the maximal jump height. Participants were permitted to perform a countermovement prior to jumping vertically or horizontally.”

I suggest to provide the sample size calculation and formula used.

Thank you. We added the following “ES = mean of the experimental group – mean of the control group/ standard deviation of the control group”

Lines 78-80: I suggest to change “were used “ in “was applied”.

Change done.

Statistic:

I suggest to carry out the correlation between anthropometric measures and jump data, and the minimum detectable change (MDC). These results could improve discussion and conclusions

We appreciate this comment; however, we did not assess body composition and therefore we were not able to determine lean body mass and fat mass, that will be more accurate. Consequently, we were not able to perform the correlation analysis. While MDC is a very interesting analysis, we did not use it because we compared two different cohorts over time and not experimental/longitudinal design to identify reliable changes (Haley y Fragala-Pinkham, 2006; Darter et al., 2013).

Darter, B. J., Rodriguez, K. M., & Wilken, J. M. (2013). Test–Retest Reliability and Minimum Detectable Change Using the K4b2: Oxygen Consumption, Gait Efficiency, and Heart Rate for Healthy Adults During Submaximal Walking. Research Quarterly for Exercise and Sport, 84(2), 223–231. doi:10.1080/02701367.2013.784720 

Haley S. M., & Fragala-Pinkham M. A. (2006). Interpreting change scores of tests and measures in physical therapy. Physical Therapy, 86(5), 735–743

Results:

Lines 84-85: Why the sample size is different? I suggest to motivate this.

The sample consisted of the boys (10-11 years of age) enrolled in primary school in 1969 (n=140) and 2016 (n = 113), and as such the sample size for each year was different

Line 92: I suggest to change “Secular”

Changed to secular.

Discussion:

Lines 111-112: “However, the 2016 boys were shorter with a lower BMI”; I think BMI is incorrect. Do you mean mass?

Thank you. We edited the text.

Lines 127-128: Did you collect data about physical activities such as IPAQ, GPAQ, questionnaires? I think these data could give an important contribution to better explain the findings.

Thank you. While physical activity data could have provided interesting insights, these questionnaires were not available in 1969.

Lines 137-145: I think that another study limit is that lean and fat mass were not assessed. I suggest to add this thought to limit session.

Thank you. This has been added to the study limitations as follows: “Assessment of body composition in order to differentiate between lean body mass and fat mass was not performed”

Finally, it’s possible that height influenced standing long jump test? I think that aspect should be rationalize/elucidate in the discussion.

Thank you. We made the following edit to the text, “Our results show that the 2016 cohort was shorter and their SLJ performance was significantly lower than in 1969. In young adults, SLJ performance has been found to be influenced by several factors including anthropometrics [20]. Taller subjects have a higher center of gravity and longer leg lever that can produce greater mechanical jump forces. It has been reported that femur length has a significant influence on VJ performance [21]. In 10 to 12 year old children, a positive moderate correlation between femur length and SJL and a weak positive correlation between standing height and SLJ was reported [22]. Given these observations, participant’s anthropometrics may effect take-off angle during the SLJ and, consequently, jump distance”.

  1. Wakai, M.; Linthorne, N.P. Optimum take-off angle in the standing long jump. Hum. Mov. Sci. 2005, 24, 81–96, doi:10.1016/j.humov.2004.12.001.
  2. Sharma, H.B.; Gandhi, S.; Meitei, K.K.; Dvivedi, J.; Dvivedi, S. Anthropometric basis of vertical jump performance: A study in young Indian national players. J. Clin. Diagnostic Res. 2017, 11, CC01–CC05, doi:10.7860/JCDR/2017/23497.9290.
  3. Sidhu, J.S. Physical attributes as indicator of performance for broad jumping. Int. J. Curr. Res. Rev. 2018, 10, 22–25, doi:10.7324/ijcrr.2018.1034.

Table 1.

I suggest to provide a title table.

Thank you. We added a tile to table 1. Participants demographics and lower body muscle power over a 47-year-period in Spanish Boys.

Round 2

Reviewer 2 Report

Dear Journal of Functional Morphology and Kinesiology Staff,
thank you very much for the possibility to re-serve as a Reviewer in a prestigious periodical like Journal of Functional Morphology and Kinesiology.

About this paper titled “A 47-year comparison of lower body muscular power 2 in Spanish boys. A short report”, I appreciate authors efforts but they reply partially to my questions, so I have new and old requests.

Lines 67: “Data were collected from two separate cross-sectional study”. I suggest to cite the first study (1969). Thank for the clarification, but you did not provide a citation. 

Lines 84-85: Why the sample size is different? I suggest to motivate this. I appreciate a more exhaustive explanation. Why the sample size in 2016 was minor.

Lines 86-87: Why you choose non-parametric test for non-normal data?

Line 92: I suggest to change “Secular” - Changed to secular. I suggest to the authors to change the word Secular because I think it’s inappropriate. From 1969 to 2016 not a century has passed.

Lines 127-128: Did you collect data about physical activities such as IPAQ, GPAQ, questionnaires? I think these data could give an important contribution to better explain the findings. Thank you. While physical activity data could have provided interesting insights, these questionnaires were not available in 1969.

I suggest to the authors to add this assumption to the study limitation.

Author Response

Dear Editor, the authors appreciate the comments for our manuscript jfmk-880289. Our manuscript has been improved with the reviewer’s comments. Changes in our manuscript have been highlighted in red in order to facilitate the location of the suggested changes. We have addressed all reviewer’s concerns and included a point-by-point response below:

REVIEWER 2:

About this paper titled “A 47-year comparison of lower body muscular power 2 in Spanish boys. A short report”, I appreciate authors efforts but they reply partially to my questions, so I have new and old requests.

Lines 67: “Data were collected from two separate cross-sectional study”. I suggest to cite the first study (1969). Thank for the clarification, but you did not provide a citation. 

The data from the 1969 cohort were not published; we received permission to use these data for the present study.

Lines 84-85: Why the sample size is different? I suggest to motivate this. I appreciate a more exhaustive explanation. Why the sample size in 2016 was minor.

Our study was carried in the same school during regular physical education classes in 1968/1969 and in 2015/2016; consequently, the sample sizes are different due to enrollment in physical education at this school during each study period of 1968/1969 and 2015/2016. Eligible participants included 10-11 year old boys who were enrolled in physical education and completed the fitness tests.

Lines 86-87: Why you choose non-parametric test for non-normal data?

Thank you. Our sample is greater than n=50 and therefore we use a Kolmogorov-Smirnov test to assess the normal distribution of the data. For normal data we used a two-tailed t-student test and for non-normal data we used Mann-Whitney  test.

We made the following edit for clarification. “A Kolmogorov-Smirnov test was conducted to determine whether data met assumptions of normality of distribution. When the data met the normal distribution, two-tailed t-tests were applied for normally distributed data and the Mann-Whitney test for non-normal data was applied to determine differences between cohorts.”

Line 92: I suggest to change “Secular” - Changed to secular. I suggest to the authors to change the word Secular because I think it’s inappropriate. From 1969 to 2016 not a century has passed.

Done. A decline trend -now line 102-

Lines 127-128: Did you collect data about physical activities such as IPAQ, GPAQ, questionnaires? I think these data could give an important contribution to better explain the findings. Thank you. While physical activity data could have provided interesting insights, these questionnaires were not available in 1969.

I suggest to the authors to add this assumption to the study limitation.

We made the following edit to clarify the limitations section.

“Additionally, we did not assess biologic maturation of the participants and we did not measure their current level of physical activity with validated questionnaires. “